# Future Prediction of Shuttlecock Trajectory in Badminton Using Player’s Information

**DOI:** 10.3390/jimaging9050099

**Published:** 2023-05-11

**Authors:** Yuka Nokihara, Ryo Hachiuma, Ryosuke Hori, Hideo Saito

**Affiliations:** Graduate School of Science and Technology, Keio University, Yokohama 223-8852, Japan; nokihara-y2896@keio.jp (Y.N.); ryo-hachiuma@keio.jp (R.H.); hori-rysk@keio.jp (R.H.)

**Keywords:** trajectory prediction, sports analysis, time-series model

## Abstract

Video analysis has become an essential aspect of net sports, such as badminton. Accurately predicting the future trajectory of balls and shuttlecocks can significantly benefit players by enhancing their performance and enabling them to devise effective game strategies. This paper aims to analyze data to provide players with an advantage in the fast-paced rallies of badminton matches. The paper delves into the innovative task of predicting future shuttlecock trajectories in badminton match videos and presents a method that takes into account both the shuttlecock position and the positions and postures of the players. In the experiments, players were extracted from the match video, their postures were analyzed, and a time-series model was trained. The results indicate that the proposed method improved accuracy by 13% compared to methods that solely used shuttlecock position information as input, and by 8.4% compared to methods that employed both shuttlecock and player position information as input.

## 1. Introduction

Recently, computer vision technologies have been employed to automate the analysis of video clips from net sports matches, such as tennis, volleyball, and badminton. These technologies enable player pose detection [1] and ball detection/tracking [2], allowing for the extraction of crucial information from match videos. This information can then be used to determine the high-level context, such as the players’ actions, during the match.

Predicting the future movement of the ball, shuttlecock, and opponent is a critical aspect of sports video analysis. The ability to forecast these movements during a rally can give players a significant advantage over their competitors. Many players rely on their experience to make these predictions, and in fast-paced sports such as badminton, being able to anticipate the movement of the shuttlecock even a fraction of a second ahead can mean the difference between winning and losing the match. Performance analysis of players can also be considered so that players can understand strategies in matches. If a player’s performance aligns with the prediction made by the proposed model, it implies that the shuttlecock’s trajectory may be easily predicted by the competing player in the match. Players aim to play unpredictably to win the match.

Most research focused on predicting future movements in net sports has centered around predicting the landing point of the ball or shuttlecock [3,4,5,6,7,8]. However, in badminton, the shuttlecock must be hit without bouncing and from a higher, faster forward position, making the prediction of the landing point insufficient for gaining an advantage in the game. To achieve this, it is necessary to predict the shuttlecock’s trajectory. Currently, trajectory prediction studies are limited to short-term events, such as the serve in table tennis or the toss in volleyball, and have not yet been applied to rallies in badminton.

This paper presents a method for predicting the future trajectory of the badminton shuttlecock during a match. One of the simple methods utilizes sequential models, such as recurrent neural networks (RNNs) [9], to model the motion of the shuttlecock by inputting previous shuttlecock trajectories and outputting future shuttlecock trajectories. To better reflect a player’s decision-making process during a match, the proposed method also considers the player’s position and posture information in addition to the shuttlecock’s position information.

The proposed method was evaluated using the shuttlecock trajectory dataset [10] and demonstrated improved accuracy compared to methods that utilized only shuttlecock position information or methods that employed both shuttlecock and player position information. The results of the study can be summarized as follows:This is a pioneering study on predicting the trajectory of the badminton shuttlecock during a match.The proposed method predicts the shuttlecock’s trajectory by considering the player’s position and posture information, in addition to the shuttlecock’s position information.The results of the experiments show that the proposed method outperforms previous methods that use only shuttlecock position information as input or methods that use both shuttlecock and player position information as input.

## 2. Materials and Methods

### 2.1. Related Work

#### 2.1.1. Future Predictions in Net Sports

Recently, research on future prediction in various net sports, such as tennis, volleyball, table tennis, and badminton, has been growing in popularity. This section introduces previous research in the field and compares them with the current research. As shown in Table 1, previous research on future prediction in net sports has focused on predicting shot direction, landing point, stroke, and trajectory. A more detailed discussion of previous research in each sport is presented below.

In tennis, Shimizu et al. [11] were the first to predict the future shot direction in three categories: right cross, left cross, and straight. They did so by using the player’s continuous position and posture information up until the moment the ball was hit. They also developed a new dataset with shot directions for their study. However, in badminton, predicting only the direction is insufficient, as players have different movements for low and high trajectories.

In table tennis, Wu et al. [4,5] predicted the landing point of the service by using the player’s motion information up until the moment just before hitting the ping-pong ball. This information was obtained through pose estimation. In volleyball, Sato et al. [6] predicted the landing point of the ball by using its velocity and 3D position, with an average error of about 0.3 m and about 1.5 s before the ball hit the floor. In badminton, Waghmare et al. [3] calculated the speed and direction of the shuttlecock to predict its landing point, using a two-dimensional laser scanner. While these methods help the player reach the landing point more quickly, they are not enough to give the player an advantage in the game by allowing them to hit the shuttlecock back faster and higher.

Stroke prediction involves predicting both the shot type and landing point. In tennis, Fernando et al. [7] predicted strokes by using a semi-supervised generative adversarial network (SGAN) [14], which combined a memory model with the automatic feature learning capabilities of a deep neural network. In badminton, Wang et al. [8] utilized a network called shuttleNet to predict the next stroke based on the current stroke. This was the first study to address stroke prediction in sports. While stroke prediction provides a better prediction of the shuttlecock’s trajectory than landing point prediction, it still falls short compared to trajectory prediction.

In volleyball, Suda et al. [12] predicted the trajectory of the toss 0.3 s before it actually occurred, using the setter player’s 3D joint positions. In table tennis, Lin et al. [13] predicted the trajectory of a subsequent serve by using a dual-network method. The method involved learning two separate parabolic trajectories: one from the service point to the landing point on the table (parabola 1) and one from the landing point to the hitting point (parabola 2). While trajectory prediction has been studied in both volleyball and table tennis, it has not yet been adequately studied in badminton.

All of the previous research has been conducted in recent years, and research on future predictions in net sports is still in its developmental stage. While shot direction prediction in tennis by Shimizu et al. [11] and stroke prediction in badminton by Wang et al. [8] have been explored, trajectory prediction has not been thoroughly studied, particularly in the case of badminton. Therefore, the aim of the current study is to predict the shuttlecock’s trajectory in badminton.

#### 2.1.2. Object Detection

Object detection is the task of identifying objects with specific attributes within an image or video and determining their location and extent by surrounding them with a bounding box.

There are two types of deep learning-based object detection methods: one-stage methods [15,16,17] that directly detect the target object from the input image, and two-stage methods [18,19,20] that first identify candidate regions in the input image and then perform detailed detection for each region. One-stage methods, such as YOLOv4 [17], prioritize processing speed and are suitable for real-time applications. Two-stage methods, such as Region-CNN (R-CNN) [18], Fast R-CNN [19], and Faster R-CNN [20], may have a slower processing speed compared to one-stage methods, but they offer higher detection accuracy. For this study, Faster R-CNN is used, as it is one of the highest-performing two-stage methods and provides more accurate information about the player’s position and posture.

#### 2.1.3. Pose Estimation

Pose estimation is the task of identifying information about a person’s posture (eyes, nose, limbs, etc.) from an image or video showing a person. The joint points of a person are obtained as keypoints.

There are two types of deep learning-based pose estimation methods: the top-down method [1,21,22,23,24,25,26,27], which performs pose estimation for each object identified after object detection in the image, and the bottom-up method [28,29,30,31,32,33], which first performs pose estimation for all objects in the image by connecting each keypoint to other objects of the same type. The former method tends to be more accurate because it estimates the posture of each object one by one, while the latter method is less accurate because it is challenging to learn to connect keypoints between the same objects. In this method, object detection and pose estimation are independent of each other in the top-down method, and HRNet [1] is applied to pose estimation after object detection.

### 2.2. Method

#### 2.2.1. Overview

This paper proposes a method for predicting the trajectory of a shuttlecock over the next (future) $n_{f}$ frames, based on the shuttlecock position, the positions, and postures of the two competing players detected during the previous (past) $n_{p}$ frames. The proposed method consists of two modules: a detection and pose estimation module for past frames, and a time series model module for future prediction, as illustrated in Figure 1.

The first module takes a sequence of past frames from badminton match videos as input and performs shuttlecock detection, player detection, and player pose estimation for each image.

In the second module, the dimensions of the shuttlecock position, player position, and player posture information are each aligned in two dimensions. The six-dimensional information is then combined and used as input for trajectory prediction.

#### 2.2.2. Detection and Pose Estimation from Past Frames

The shuttlecock position information is obtained from a shuttlecock detector, such as TrackNet [34]. These are two-dimensional coordinates $x_{s}$ and $y_{s}$ on the image.

For player detection and pose estimation, the MMPose framework [35] is utilized, which includes several pose estimation models and pre-trained models. An object detection model detects players and records their positions as bounding boxes, while a pose estimation model estimates the posture of the players and records their keypoints.

To detect humans, an object detector is utilized; we employ Faster R-CNN [20] trained on the Microsoft Common Objects in Context (MS COCO) dataset [36]. The detector provides bounding boxes and confidence scores as detection results. Bounding boxes are represented by four two-dimensional coordinate points on the image when a human is enclosed by a rectangle. The confidence score ranges from 0 to 1 and indicates the likelihood that the object within the detected bounding box is a human. By using the confidence score, only players, not referees or spectators, are detected. Referees and spectators that are detected alongside players have a lower confidence score, as they may be sitting, have only their faces in the image, be facing sideways, or appear small. Thus, the person with the highest confidence score is identified as the player, and the player’s bounding box is obtained, as shown in Figure 2.

The player position information is obtained by calculating the normalized center coordinates $\left(x_{player},y_{player}\right)$ of the player with each coordinate in Figure 3 as follows:(1)$$\left(x_{player},y_{player}\right)=\left(\frac{({bbox}_{left}+{bbox}_{right})/2}{{img}_{width}},\frac{({bbox}_{top}+{bbox}_{bottom})/2}{{img}_{height}}\right).$$

The players are distinguished by assigning a number to each of them in the image, starting with the player on the lower side. The player position information is represented as four-dimensional coordinates $\left(x_{p1},y_{p1},x_{p2},y_{p2}\right)$, where $x_{p1}$ represents the *x* coordinate of the player who is shown on the lower side of the image.

After the player detection, HRNet [1], which has been trained by MS COCO, is used as the pose estimator. As shown in Figure 4, 17 joints (eyes, ears, nose, shoulders, elbows, wrists, hips, knees, and ankles) are detected as keypoints in the bounding box obtained from player detection.

The coordinate values are expressed as absolute coordinates on the image. For example, if the coordinates of a keypoint are $\left(x_{k},y_{k}\right)$ on the image, the normalized absolute coordinates $\left({ax}_{k},{ay}_{k}\right)$ of the keypoint are calculated using the values in Figure 3 as follows:(2)$$\left({ax}_{k},{ay}_{k}\right)=\left(\frac{x_{k}}{w},\frac{y_{k}}{h}\right),$$
where w, and *h* are the width and height of the input image, respectively. The posture information of the two players is represented as a 68-dimensional feature vector $\left(x_{p1k1},y_{p1k1},\ldots ,x_{p2k17},y_{p2k17}\right)$, where $x_{pikj}$ represents the *x* coordinate of the first keypoint of the player who is shown on the lower side of the image.

#### 2.2.3. Time Series Model for Future Prediction

The proposed method predicts the trajectory by inputting three types of information: the shuttlecock position information obtained by the shuttlecock detector, the player position information obtained by the object detector, and the player posture information obtained by the pose estimation.

At this point, the shuttlecock position information is 2-dimensional, the player position information is 4-dimensional, and the player posture information is 68-dimensional. In order to make three types of information into a feature vector of the same space, each of them is embedded into two dimensions by a fully connected layer as shown in Table 2. The shuttlecock position information and the player position information and the player posture information are combined to form a 6-dimensional feature vector.

The combined information is fed into a long short-term memory (LSTM) network [37], which is the second module used for predicting the shuttlecock trajectory. The LSTM network uses multiple past inputs stacked together as its input, and its output is then forwarded to the fully connected layer for further processing.

### 2.3. Experiment

#### 2.3.1. Dataset

The shuttlecock trajectory dataset [10] was utilized, which was created for training and testing the TrackNet [34] and TrackNetV2 [38] models for badminton applications. This dataset comprises 26 match videos for training and 3 match videos for testing. The match videos have a resolution of 1280×720 and a frame rate of 30 fps, and are separated by rallies, which refer to recordings that start with a serve and end with a score.

Each frame in the dataset provides information about the shuttlecock’s position and the moment it hits the racket. Twenty-three matches were used for training, excluding amateur matches, while three matches were used for testing. The professional matches used in this study were singles matches held in international tournaments between 2018 and 2021. The 23 match rally videos for training were randomly split into an 80% training set and a 20% validation set. The three-match rally videos for testing were used as the test set.

Data cleansing was carried out on the dataset to enhance prediction accuracy. The position coordinates of the shuttlecock in the dataset were set to (0,0) whenever it was hidden by a person or not visible, which could negatively affect learning since the shuttlecock would appear to move unnaturally in the frames before and after it. Frames with such issues were removed from the dataset to ensure that consecutive frames with unnatural shuttlecock movements were not used for learning.

Data augmentation was also carried out to increase the diversity of the training data. Since flipping a badminton match video upside down would no longer be appropriate for the sport, only left–right flipping and translations were performed. The original image was flipped left and right with a probability of 50% and then translated to the right or bottom in a range of 0 to 50 pixels relative to the width and height of the image. Figure 5 shows the results of the data augmentation.

#### 2.3.2. Evaluation Metrics

Two types of displacement errors, namely the average displacement error (ADE) and final displacement error (FDE), were used as evaluation metrics in this experiment. ADE is the average of errors across all output frames, while FDE is the error at the final point of the output trajectory. The Euclidean distance was calculated using the two-dimensional coordinates in a 1280×720 pixel image, and the unit of measurement is pixels. ADE is more important in this task since it predicts the trajectory for multiple frames rather than just the landing point.

#### 2.3.3. Network Training

We implemented the proposed method in PyTorch [39] (1.12.1+cu102, with Python 3.7.13) and ran it on the NVIDIA TITAN RTX processing unit using CUDA 11.4. For LSTM, the number of layers was set to 3, the hidden layer to 128 dimensions, and the network was optimized using Adam [40], with a weight decay of $10^{-4}$, the momentums $\beta_{1}=0.5$ and $\beta_{2}=0.999$, and a learning rate of 0.02. The model was trained for 400 epochs with four input frames and 12 output frames for all cases.

The mean squared error (MSE) was employed as the loss function:(3)$$Loss=\frac{1}{D}\sum_{i=1}^{D}{\left({F(x)}_{i}-{gt}_{i}\right)}^{2},$$
where *F* is the time-series model LSTM of the proposed method, *x* is its input data and gt is the ground-truth data of the output shuttlecock position. F(x) and gt are *D*-dimensional vectors and ${F(x)}_{i}$ is the value of F(x) on the *i* dimension. The output results were compared with the ground-truth data to calculate the error. Then, the parameters were updated to reduce the error using backpropagation.

#### 2.3.4. Other Models

##### Baseline Models

To verify the effectiveness of the proposed method, two baseline models were set up. One is to input only the shuttlecock position information without any player information, and the other is to input the shuttlecock and player position information without any player posture information.

##### Other Time-Series Models

LSTM was used as the model for trajectory prediction, but four other time-series models were also examined: an RNN [9], gated recurrent unit (GRU) [41], Transformer [42] and sequence-to-sequence learning with neural networks (Seq2Seq) [43]. The network parameters were set the same as in the proposed method model for all time-series models, except for the learning rate. Only the learning rate was set as shown in Table 3.

##### Other Representations of Posture Information

In addition to the absolute coordinate values on the image used in this method, we also examined other ways of representing the posture of the players: the relative joint positions in the bounding box and the heat map generated for the pose estimation.

If the coordinates of a keypoint are $\left(x_{k},y_{k}\right)$ on the image, the normalized relative coordinates $\left({rx}_{k},{ry}_{k}\right)$ to the bounding box are calculated using the values in Figure 3 as follows:(4)$$\left({rx}_{k},{ry}_{k}\right)=\left(\frac{x_{k}-{bbox}_{left}}{{bbox}_{right}-{bbox}_{left}},\frac{y_{k}-{bbox}_{top}}{{bbox}_{bottom}-{bbox}_{top}}\right).$$

The accuracy of three inputs to the time-series model was compared: absolute coordinate values on the image, relative coordinate values to the bounding box, and heatmap generated by pose estimation, respectively.

When using absolute coordinates on the image, the spatial coordinates are the same as the position information of the shuttlecock and players. When using relative coordinates to the bounding box, the spatial coordinates are different from the position information of the shuttlecock and players. Relative coordinates have a more significant influence on the posture information than absolute coordinates because the change in posture is greater with relative coordinates. Figure 6a shows the overview in the case of using the joint positions to represent the postures of the players.

When using the heatmap, a heatmap was first generated for each of the two players in the pose estimation part and superimposed, as shown in Figure 7. Then, the heatmap including the posture information for the two players was sent to ResNet-18 [44] to extract a 512-dimensional feature vector, which was input to the time-series model together with the shuttlecock position and the players’ positions. Figure 6b shows the overview in the case of using the heatmap to represent the postures of the players.

#### 2.3.5. Data Augmentation Values

Experiments were conducted for data augmentation with different probabilities of left–right flipping and ranges of parallel shifts. The probability of left–right reversal was set to 0%, 25%, 50%, or 75%, where 0% means that the image is not flipped. The range of translation was set to 0 pixels, 50 pixels, or 100 pixels, where 0 means no translation.

#### 2.3.6. The Number of Frames of Past/Future

Experiments were conducted with the number of past and future frames fixed at 4 and 12, respectively. For the results shown in Section 3.3, five different combinations of the number of past and future frames were tested by increasing the number of past frames from 4 to 12, while keeping the total number of past and future frames at 16.

## 3. Results

### 3.1. Comparison with Other Models

As for the quantitative evaluation, the results of comparing this method with the baseline methods and the methods using other time-series models are shown in Table 4 and Table 5. As a qualitative evaluation, the results of comparing this method with the two baseline methods are shown in Figure 8, and the results of comparing this method with methods using other time-series models are shown in Figure 9.

The proposed method shows the best results for ADE and FDE. It also qualitatively predicts the trajectories closest to the ground truth.

Compared to the method with the baseline methods, the proposed method improves ADE by about 13% compared to the method using only the shuttlecock position information as input, and by about 8.4% compared to the method using the shuttlecock and player position information as input. This shows that the proposed method effectively uses all the information on the shuttlecock position, player position, and player posture. Compared to the method with the highest accuracy for each time-series model, the proposed method using LSTM improves ADE by about 9.8%, 5.0%, 20%, and 12% compared to the methods using RNN, GRU, Transformer, and Seq2Seq, respectively. Therefore, it is shown that the proposed method using LSTM is the best model among the time-series models considered.

### 3.2. Comparison by Data Augmentation Values

The results of the experiments with different probabilities of left–right flipping and different ranges of translation are shown in Table 6 and Table 7. The proposed method shows the best results for ADE and FDE. This indicates that a 50% probability of left–right flipping and a 50-pixel range of translation are appropriate among those considered in this study, and that the data augmentation is effective.

### 3.3. Comparison of the Number of Frames of Past/Future

The accuracy improves as the number of input frames increases and the number of frames to be predicted decreases as shown in Figure 10, Table 8 and Table 9.

## 4. Discussion and Conclusions

### 4.1. Limitation

This study proposed a method for predicting future trajectories in badminton match videos. However, several limitations exist in the framework.

The first limitation is the position of the cameras. The dataset used in this study includes high-level tournaments, such as the World Championships, all of which were recorded from a distance so that the entire court could be seen. This makes it easy to obtain information on shuttlecocks and players. However, recording from a distance requires equipment, such as a tripod and a photographer, making it difficult to set up a camera when attending a match alone or when there is no access to a suitable recording location. In addition, since the camera is recording from a distance, it is likely that people may pass in front of the camera. If there are many frames where the shuttlecock is not in the angle of view and cannot be tracked, it would be difficult to predict.

Second is the adaptation to sudden changes in trajectory. As shown in Figure 11, if the trajectory changes at the moment of hitting back or the moment the shuttlecock touches the floor, it is quite difficult to predict. Even if it could be predicted that the trajectory would change after bouncing back, the direction of the change could be wrong. In this method, the position and posture information of the player was input in addition to the shuttlecock’s position information, which enabled some frames to adapt to changes in trajectory, but there were still many frames that were difficult to predict.

### 4.2. Future Work

There are two things that need to be worked on in the future to make this research work with in-the-wild data.

The first is to improve the generalization performance of the model. For the practical scenario, it is necessary to confirm that the model is general enough by testing the proposed model on multiple data sets in badminton (e.g., those with different camera locations and those containing matches of amateur players) and other net sports, such as tennis and table tennis. For this purpose, it would also be useful to be able to automate the rally segmentation and shuttlecock position detection in match videos.

The second is to perform trajectory prediction in 3D space. Since this research is based on the novel task of performing trajectory prediction in badminton, we performed the prediction in 2D space as a first step. If we can further develop this task to predict in 3D space and project the predicted trajectory into real space, it can be used for training to predict the trajectory of a shuttlecock hit by an opponent or to prevent an opponent from predicting the trajectory of a shuttlecock hit by oneself. This can further help players improve their skills.

### 4.3. Conclusions

This paper approached the novel task of predicting the trajectory of the shuttlecock in a badminton match video and proposed a trajectory prediction method that uses information about the shuttlecock’s position and the players’ positions and the players’ postures. Experiments comparing the proposed method with the baseline method were conducted to confirm the effectiveness of the proposed method. Furthermore, experiments with different time-series models show that the LSTM used in this method achieves the highest accuracy.

## Figures and Tables

**Figure 1 jimaging-09-00099-f001:**
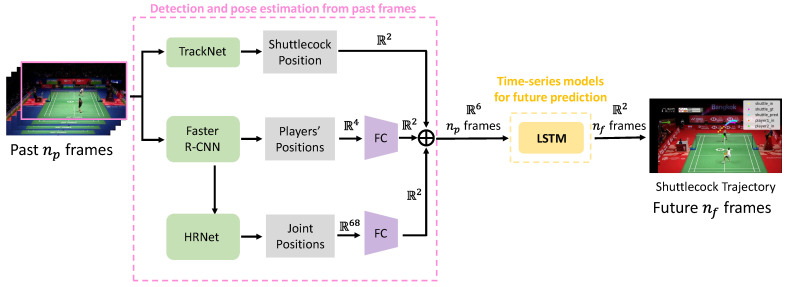
Overview of the proposed method.

**Figure 2 jimaging-09-00099-f002:**
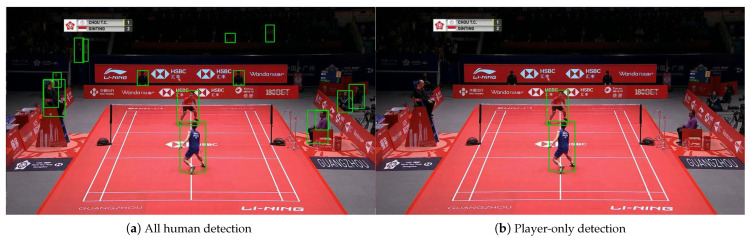
Player detection.

**Figure 3 jimaging-09-00099-f003:**
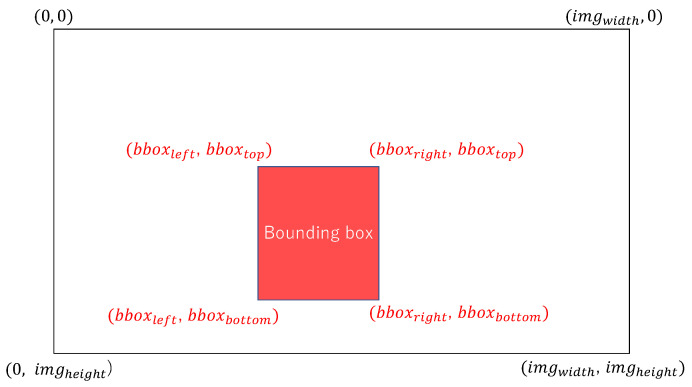
Coordinates of image and bounding box.

**Figure 4 jimaging-09-00099-f004:**
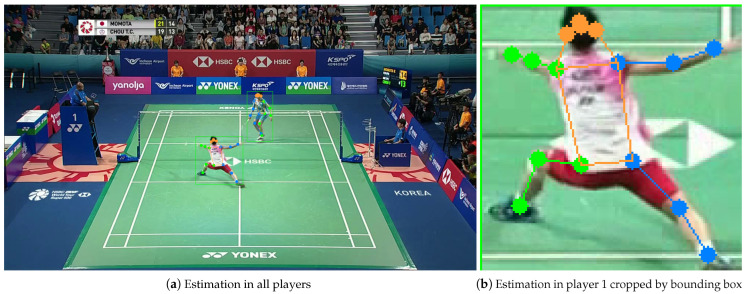
Pose estimation.

**Figure 5 jimaging-09-00099-f005:**
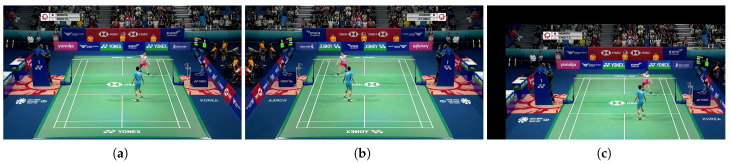
Data augmentation. (**a**) Original image, (**b**) image after left–right flipping, and (**c**) image with 100 translations to the right and 100 translations to the bottom.

**Figure 6 jimaging-09-00099-f006:**
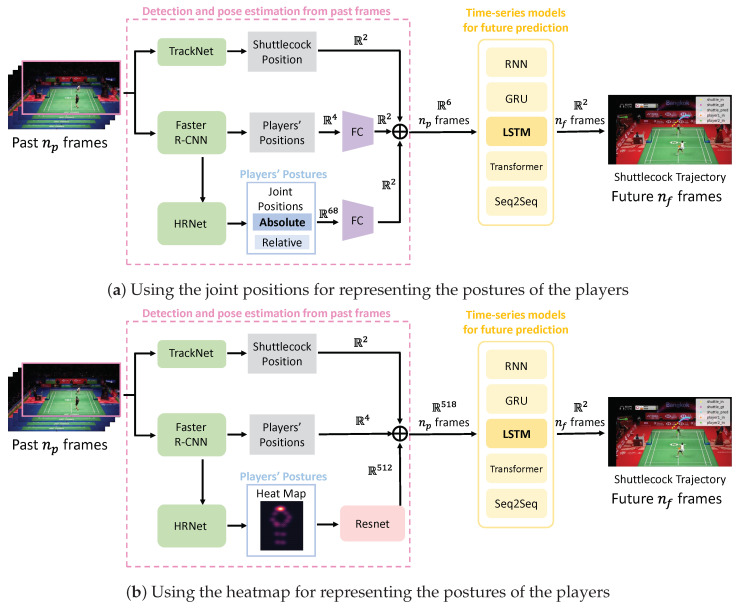
Other time-series models and other way of representing the players’ posture.

**Figure 7 jimaging-09-00099-f007:**
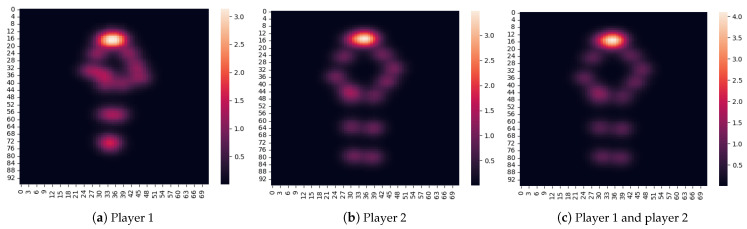
Heatmap. (**c**) is a heat map for two players superimposed on (**a**,**b**).

**Figure 8 jimaging-09-00099-f008:**
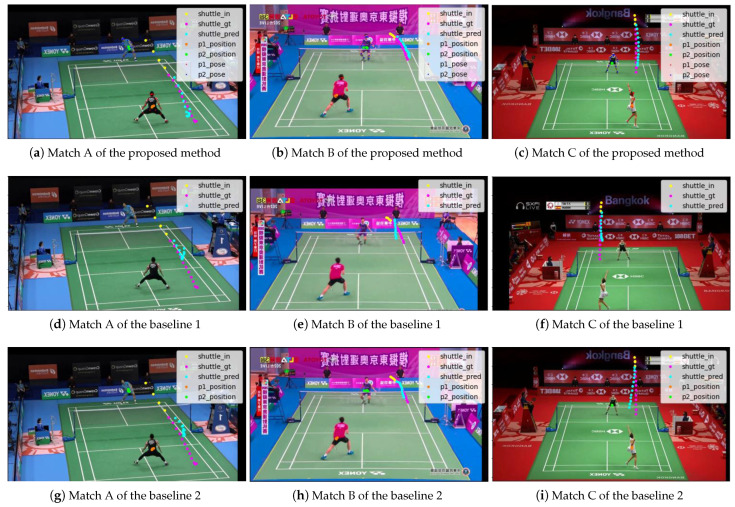
Future predictions of shuttlecock trajectories compared to baseline 1 and baseline 2.

**Figure 9 jimaging-09-00099-f009:**
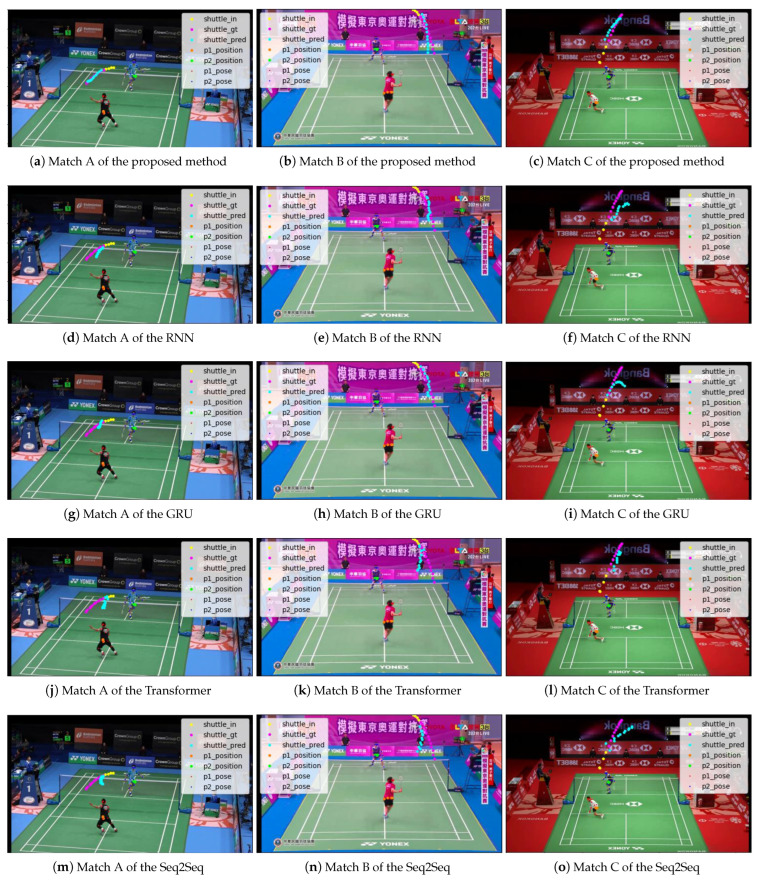
Future predictions of shuttlecock trajectories compared to other models.

**Figure 10 jimaging-09-00099-f010:**
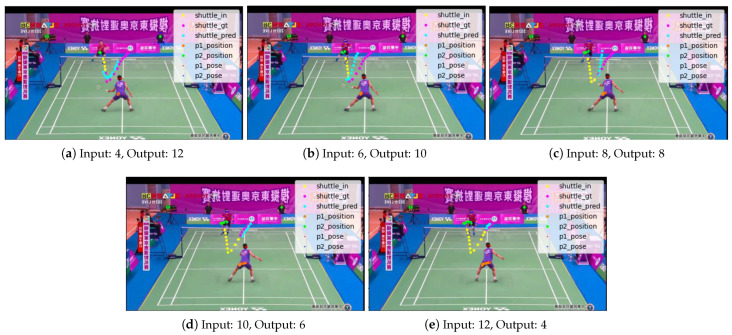
Comparison according to the number of frames of input/output.

**Figure 11 jimaging-09-00099-f011:**
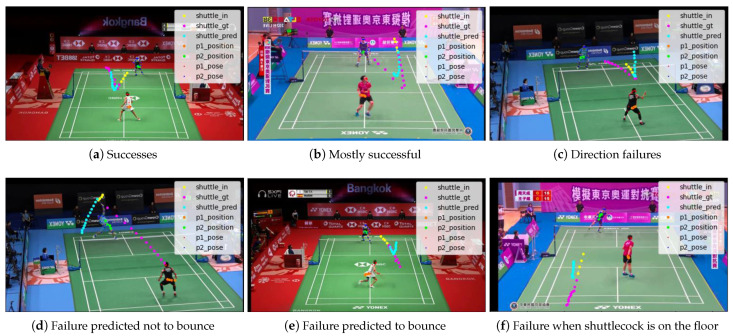
The adaptation to sudden changes in trajectory.

**Table 1 jimaging-09-00099-t001:** Previous research of future predictions in net sports.

Year	Author	Sports	Prediction Target
2019	Shimizu et al. [11]	Tennis	Shot direction
2016	Waghmare et al. [3]	Badminton	Shuttlecock landing point
2019, 2020	Wu et al. [4,5]	Table tennis	Serve landing point
2019	Sato et al. [6]	Volleyball	Ball landing point
2019	Fernando et al. [7]	Tennis	Stroke
2022	Wang et al. [8]	Badminton	Stroke
2019	Suda et al. [12]	Volleyball	Toss trajectory
2020	Lin et al. [13]	Table tennis	Serve trajectory
2022	Proposed	Badminton	Shuttlecock trajectory

**Table 2 jimaging-09-00099-t002:** Number of dimensions.

Target	Original	After Embedding
Shuttlecock position	2	2
Player position	4	2
Player posture	68	2
All	74	6

**Table 3 jimaging-09-00099-t003:** Learning rate for each time-series model.

Models	Learning Rate
RNN	0.005
GRU	0.01
Transformer	0.001
Seq2Seq	0.02

**Table 4 jimaging-09-00099-t004:** ADE results. The last row shows the result of the proposed method. The alphabet of the player posture represents the input method to the time-series model: A is the absolute coordinates on the image, R is the relative coordinates in the bounding box, and H is the heat map. Bold font indicates the best (the minimum) ADE.

Input Data			Models	ADE (Pixel)			
Shuttlecock Position	Player Position	Player Posture		Match A	Match B	Match C	Average
∘	-	-	LSTM	54.1	40.7	45.8	47.6
∘	∘	-	LSTM	51.6	36.5	43.9	45.2
∘	∘	R	RNN	52.0	38.9	44.9	45.9
∘	∘	H	RNN	63.4	46.0	60.2	58.4
∘	∘	A	RNN	55.4	38.0	49.8	48.8
∘	∘	R	GRU	48.9	36.1	42.9	43.6
∘	∘	H	GRU	59.8	42.8	52.9	51.1
∘	∘	A	GRU	53.4	37.0	43.5	45.4
∘	∘	R	Transformer	59.3	44.5	49.0	51.5
∘	∘	H	Transformer	81.0	65.7	73.4	74.3
∘	∘	A	Transformer	60.3	44.2	47.4	51.6
∘	∘	R	Seq2Seq	**40.4**	53.8	45.8	47.3
∘	∘	H	Seq2Seq	53.8	40.7	45.9	47.7
∘	∘	A	Seq2Seq	56.7	46.6	43.0	48.9
∘	∘	R	LSTM	46.5	34.4	41.0	**41.4**
∘	∘	H	LSTM	56.4	44.4	51.9	51.3
∘	∘	A	LSTM	48.3	**33.0**	**39.1**	**41.4**

**Table 5 jimaging-09-00099-t005:** FDE results. The last row shows the result of the proposed method. The alphabet of the player posture represents the input method to the time-series model: A is the absolute coordinates on the image, R is the relative coordinates in the bounding box, and H is the heat map. Bold font indicates the best (the minimum) FDE.

Input Data			Models	FDE (Pixel)			
Shuttlecock Position	Player Position	Player Posture		Match A	Match B	Match C	Average
∘	-	-	LSTM	94.1	74.1	75.0	81.5
∘	∘	-	LSTM	88.0	63.4	70.9	76.1
∘	∘	R	RNN	90.1	66.7	74.5	78.0
∘	∘	H	RNN	108.7	77.2	101.1	98.7
∘	∘	A	RNN	94.0	64.1	74.5	79.1
∘	∘	R	GRU	87.4	67.6	69.8	76.2
∘	∘	H	GRU	109.3	76.7	96.0	92.9
∘	∘	A	GRU	95.8	70.0	73.5	80.3
∘	∘	R	Transformer	97.1	74.2	77.1	84.0
∘	∘	H	Transformer	128.6	95.7	109.0	113.8
∘	∘	A	Transformer	99.0	71.9	75.2	83.7
∘	∘	R	Seq2Seq	64.3	91.3	73.5	77.9
∘	∘	H	Seq2Seq	93.7	72.3	78.6	83.1
∘	∘	A	Seq2Seq	92.0	73.2	64.9	78.1
∘	∘	R	LSTM	**81.5**	61.6	68.0	71.7
∘	∘	H	LSTM	97.5	79.1	86.7	88.4
∘	∘	A	LSTM	83.1	**59.0**	**63.3**	**70.4**

**Table 6 jimaging-09-00099-t006:** ADE results from changing the probability of left–right flipping and the range of translation in augmentation.

Augmentation		ADE (Pixel)			
**Shift Range (Pixel)**	**Probability of Flip (%)**	**Match A**	**Match B**	**Match C**	**Average**
0	0	50.0	36.8	40.2	43.2
0	25	48.8	34.5	39.2	42.0
0	50	51.4	37.9	41.2	44.3
0	75	48.8	34.5	41.3	43.2
50	0	51.7	38.0	40.7	44.3
50	25	49.0	35.1	39.7	42.6
50	50	48.3	33.0	39.1	41.4
50	75	48.0	36.4	41.7	42.5
100	0	50.6	39.4	41.8	44.6
100	25	49.1	36.0	41.7	42.8
100	50	51.8	38.3	43.6	46.6
100	75	51.8	38.2	43.0	44.9

**Table 7 jimaging-09-00099-t007:** FDE results from changing the probability of left–right flipping and the range of translation in augmentation.

Augmentation		FDE (Pixel)			
**Shift Range (Pixel)**	**Probability of Flip (%)**	**Match A**	**Match B**	**Match C**	**Average**
0	0	85.7	66.7	68.1	74.7
0	25	85.2	62.5	65.9	72.8
0	50	86.8	67.2	69.0	74.7
0	75	83.7	61.1	68.3	73.4
50	0	86.7	66.8	68.0	75.0
50	25	84.1	64.5	65.9	73.3
50	50	83.1	59.0	63.3	70.4
50	75	81.7	63.6	66.5	71.3
100	0	87.6	65.6	68.1	75.0
100	25	83.2	63.3	68.6	72.2
100	50	85.7	62.8	68.5	75.3
100	75	89.0	68.2	70.1	76.3

**Table 8 jimaging-09-00099-t008:** ADE results when input/output frames are changed.

Input Frames	Output Frames	ADE (Pixel)			
		Match A	Match B	Match C	Average
4	12	48.3	33.0	39.1	41.4
6	10	42.7	31.5	35.1	37.2
8	8	31.9	25.9	29.0	29.5
10	6	22.4	22.4	25.3	23.5
12	4	19.0	19.2	18.7	18.9

**Table 9 jimaging-09-00099-t009:** FDE results when input/output frames are changed.

Input Frames	Output Frames	FDE (Pixel)			
		Match A	Match B	Match C	Average
4	12	83.1	59.0	63.3	70.4
6	10	70.8	51.5	56.7	61.0
8	8	52.0	43.9	47.2	48.0
10	6	36.1	35.8	37.8	36.7
12	4	28.5	29.8	26.9	28.1

## Data Availability

Not applicable.

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
