# Peer review of "Future Prediction of Shuttlecock Trajectory in Badminton Using Player’s Information"

_2313-433X, 2023, doi:10.3390/jimaging9050099_

Round 1

Reviewer 1 Report

The article titled "Future Prediction of shuttlecock Trajectory in Badimnton Using Player's Information" presents the resulsts of experiments with predicting the position of the shuttlecock in the camera image plane in several frames after observing previous n video frames using deep learning methods.

Overall the paper is very readable and the topic is interesting, however there are some missing details that should be added to improve the paper:

2.2.4 Trajectory prediction: please explain in more detail how exactly is embedding done from the original dimension into the 6-dimensional vector. It is not entirely clear where this fully connected layers stands in the overall model and how it is trained.

2.3.5. Please elaborate in more detail the heatmap approach, esp. the last sentence  "Then, in the trajectory prediction part, the heatmap including the posture information for the two players was feature extracted using ResNet-18 [44] and input to the time-series model together with the position information", 

Table 4.: I suggest marking the suggested method somehow to stand out

Discussion: Are there any practical applications of the proposed models?

Author Response

Thank you for your review comments.  Reviewer 1 mentions that some missing details should be added to improve the paper.  The statements and the replies from the authors are as follows:

  • 2.2.4 Trajectory prediction: please explain in more detail how exactly is embedding done from the original dimension into the 6-dimensional vector. It is not entirely clear where these fully connected layers stand in the overall model and how it is trained.
    • Figure 1 is revised to make the embedding clearly explained.  As shown in the revised Figure 1, the detected players' positions and the joint positions are connected to each fully connected layer to get R2 outputs.  They are combined with the detected shuttlecock position  R2, then inputted to LSTM as the time-series model to predict the shuttlecock trajectory in future frames.   To clarify this explanation, we revised 2.2 entirely.

  • 2.3.5. Please elaborate in more detail on the heatmap approach, esp. the last sentence  "Then, in the trajectory prediction part, the heatmap including the posture information for the two players was feature extracted using ResNet-18 [44] and input to the time-series model together with the position information",
    • The heatmap approach is one of the variations in the way of representing the players' posture to be inputted into the time-series model.  The explanation is elaborated by revising 2.3.4. Figure 7 is also added to make it clearly explained.
  • Table 4.: I suggest marking the suggested method somehow to stand out
    • The proposed method is explicitly indicated in the independent line at the bottom of Tables 4 and 5.  The captions are also revised to make the evaluation results of the proposed method presented.
  • Discussion: Are there any practical applications of the proposed models?
    • As a practical application of the proposed models, we can consider the performance analysis of the players, so that the players can understand the strategy in the matches.  If the player's performance is the same as the prediction by the proposed model, this implies that the shuttlecock trajectory may easily be predicted by the competing player in the match.  The players wish to play in an unpredicted manner to win the match.   The related discussion has already been given in the second paragraph of the first section, but we elaborate on it in the revised manuscript.

Reviewer 2 Report

The proposed work proposes an approach to the analysis of video data from badminton sports to provide insights into the performance of players so that they can improve their playing. More specifically, the goal is to forecast shuttlecock trajectory during a game to obtain an advantage over the opponent teams.

The paper is written in very good English.

The relevant literature is reviewed very well.

The novelty of this work lies in that it predicts shuttlecock trajectories taking into account player positions and postures, not only the shuttlecock position.

There is no new theory proposed but the combination of the employed theoretical components is novel. Also, the employed components are properly used.

The proposed method is quantitatively evaluated on a public benchmark dataset and comparatively evaluated against other methods.

I have only the following comments for improvement.

- The partitioning of the paper into sections is somewhat unconventional. Section 2, includes both relevant work and the definition of the proposed method. Usually, relevant work is provided in a separate section. Also, Section 2 contains an experiment which one would expect to be in the “Experiments” section.

- The examples provided could note the particular dataset and frame number(s) so that forthcoming authors can compare their results on the particular examples.

Author Response

We would like to thank the comment of Reviewer 2.  Reviewer 2 suggests the following points to improve the manuscript.

  •  The partitioning of the paper into sections is somewhat unconventional. Section 2, includes both relevant work and the definition of the proposed method. Usually, relevant work is provided in a separate section. Also, Section 2 contains an experiment which one would expect to be in the “Experiments” section.   
    • We also agree with the suggestion.  However, the editorial office of this journal first suggests adjusting the section name to the journal's template.  Therefore, we have edited to obey the template, and will not re-edit again to adapt the way of Reviewer 2 suggested.  We would appreciate it if Reviewer 2 will approve our decision, such that we will keep the current way of making the sections according to the journal's office suggestion.
  • The examples provided could note the particular dataset and frame number(s) so that forthcoming authors can compare their results on the particular examples.
    • Thank you for the suggestion, but we will not put the note because the most important result in this paper is the quantitative results shown in Tables 4,5, 6,7, 8, and 9, which are evaluated by using all data in the dataset, where the note of the frame numbers does not make much sense.  I would like to appreciate it if Reviewer 2 could understand our decision.

Round 2

Reviewer 1 Report

This is a revised version of the previously considered manuscript. The authors have addressed the issues with the last version and I recommend the article for publication.